# Botulinum Toxin Type A for Pediatric Torticollis: A Review of Clinical Research

**DOI:** 10.3390/toxins17110543

**Published:** 2025-11-01

**Authors:** Na-Yoen Kwon, Soo-Hyun Sung, Hyun-Kyung Sung

**Affiliations:** 1Department of Obstetrics and Gynecology, College of Korean Medicine, Ga-Chon University, Seongnam-si 13120, Republic of Korea; kwonnay@gachon.ac.kr; 2Department of Policy Development, National Institute for Korean Medicine Development, Seoul 04516, Republic of Korea; koyote10010@nikom.or.kr; 3Department of Education, College of Korean Medicine, Dongguk University, Gyeongju-si 38066, Republic of Korea

**Keywords:** botulinum toxin, congenital muscular torticollis, botulinum toxin type A

## Abstract

Pediatric torticollis, predominantly resulting from congenital muscular torticollis, is characterized by unilateral shortening of the sternocleidomastoid muscle, leading to head tilt and limited cervical mobility. Conventional management primarily involves physical therapy and repositioning strategies, with most infants achieving full recovery. However, a subset of patients exhibits persistent symptoms despite conservative treatment. Botulinum toxin type A (BoNT-A) has emerged as a minimally invasive adjunct intervention that targets muscular hypertonicity by inhibiting acetylcholine release at neuromuscular junctions. This scoping review synthesizes clinical evidence from six studies, including randomized controlled trials and case reports, assessing the efficacy and safety of BoNT-A in pediatric torticollis. Results indicate consistent improvements in range of motion, head posture correction, and patient satisfaction, with rare and mild adverse events such as local bruising and transient muscle weakness. Despite promising outcomes, variability in dosing, injection protocols, and follow-up durations underscores the need for standardized treatment guidelines and further high-quality research. These findings support BoNT-A as a valuable therapeutic option for refractory pediatric torticollis, warranting integration into multidisciplinary care frameworks.

## 1. Introduction

Pediatric torticollis, most commonly caused by congenital muscular torticollis, is a postural condition evident shortly after birth, characterized by lateral head tilt and rotation of the chin to the opposite side due to unilateral shortening or fibrosis of the sternocleidomastoid (SCM) muscle, sometimes accompanied by other musculoskeletal or neurological conditions [1]. This disorder typically presents within the first few months of life and manifests as a persistent head tilt toward one side with the chin rotated to the opposite direction [2]. The reported prevalence of pediatric torticollis ranges from 0.3% to 2% among newborns, making it a frequently encountered musculoskeletal condition in infants [3].

Conventional treatment for pediatric torticollis primarily involves noninvasive approaches, with physical therapy as the mainstay [1]. Standard regimens include passive stretching exercises, repositioning strategies, and sometimes the use of orthotic devices to encourage symmetrical growth and improve range of motion (ROM) [4]. Most cases respond well to these conservative interventions, especially when initiated early, and prognosis for full recovery is generally favorable; success rates approach 90–99% in infants treated with physical therapy [5]. However, in a subset of children, persistent symptoms or limited improvement after physical therapy calls for consideration of adjunctive treatments [6].

Botulinum toxin type A (BoNT-A) is a blocking agent that inhibits acetylcholine release at the neuromuscular junction, leading to temporary muscle relaxation [7]. In pediatric torticollis cases where conservative treatments are insufficient, BoNT-A injections into affected muscles such as the SCM, trapezius, or scalene muscles may facilitate correction of abnormal muscle contraction and improve neck mobility [8]. BoNT-A offers a minimally invasive alternative, and its use is supported by a generally favorable safety profile in adults; however, the safety and side effect profile in the pediatric population may differ due to ongoing neuromuscular system development. Although most reported adverse events in children are mild and transient, such as local bruising and minor neck weakness, serious events like generalized weakness or respiratory difficulties have been reported, particularly in younger or vulnerable pediatric patients. Therefore, careful dosing and monitoring are essential when using BoNT-A in children [9,10].

Existing research on BoNT-A for torticollis has largely focused on adult populations, and until recently, systematic reviews addressing BoNT-A therapy in children have been lacking. A systematic review on botulinum toxin for congenital muscular torticollis was published in 2020 [11], which performed a comprehensive literature review including both pediatric and adult subjects with congenital muscular torticollis. The purpose of this scoping review is to evaluate the clinical efficacy and safety of botulinum toxin type A in the management of pediatric torticollis, addressing an important gap in the literature and informing evidence-based care strategies for affected children.

## 2. Results

### 2.1. Study Description

A total of eight studies [12,13,14,15,16,17,18,19] were included in this review according to the predefined inclusion criteria (Figure 1). The publication years of the selected studies ranged from 2005 to 2023. In terms of study design, two randomized controlled trials (RCTs), five case reports, and one cohort study were included. The studies were conducted across four countries—the United States (*n* = 3), China (*n* = 2), South Korea (*n* = 1), and New Zealand (*n* = 1)—with one multinational study from the United States and Thailand (*n* = 1). The characteristics of the eight studies are presented in Table 1.

### 2.2. Participants

A total of 309 participants with torticollis symptoms were included eight studies [12,13,14,15,16,17,18,19]. The number of participants per study ranged from a minimum of 1 to a maximum of 134. The age of the study subjects was limited due to the focus on studies on children and adolescents under 20 years of age during the selection process, but the average age ranged from a minimum of 52 days to 14 months, and most studies included infants under 1 year of age. Most of the study subjects had tried conservative treatments such as physical therapy first, but when the treatment effect was not clear, they underwent botulinum toxin treatment. Only the study [16] clearly stated that botulinum toxin treatment was performed from the beginning.

### 2.3. Botulinum Toxin

BoNT-A administrations primarily targeted the SCM, upper trapezius, and scalene muscles via intramuscular (IM) injections. The dose per treatment session ranged from approximately 15 to 100 units (IU or U), with an injection volume of 0.5 to 1.0 mL (cc). These dosing and injection volumes were adjusted according to patient-specific factors such as age and muscle size, as well as the requirements of each study protocol (Table 2).

Most studies reported a single injection session, though some extended treatment to two sessions spaced over several months. The number of injection sites per session typically ranged from three to four, predominantly focusing on the SCM and upper trapezius, with some protocols also including scalene muscles (Table 2). Five studies [12,13,14,15,17] used products from Allergan Inc., while two studies [16,18] employed formulations manufactured by the Lanzhou Institute of Biological Products Co., Ltd. The remaining study did not specify the product used. Overall, dosing regimens reflected a personalized approach considering anatomical and clinical variability among patients.

### 2.4. Outcome Measures

The evaluation index for BoNT-A treatment was commonly used across all eight studies [12,13,14,15,16,17,18,19], with head tilt and neck rotation most frequently measured despite variations in the details of ROM assessments among studies. In addition to these, the thickness and length of the SCM muscle, as well as patient or caregiver satisfaction, were also evaluated. All studies consistently demonstrated significant improvements in symptoms following BoNT-A treatment, supporting its efficacy and safety in reducing muscle-related dysfunction.

### 2.5. Adverse Events

In the botulinum toxin treatment group, there were no side effects in two studies [15,19], and in the other five studies, side effects included bruising (*n* = 3), edema (*n* = 1), mild dysphasia (*n* = 2), neck weakness (*n* = 2), sore neck (*n* = 1), and slight weakness in sucking (*n* = 1). If these side effects are classified according to the Common Terminology Criteria for Adverse Events (CTCAE), they are all considered to be mild side effects corresponding to Grade 1. The remaining study [17] did not report any adverse events.

### 2.6. Co-Interventions

Of the eight included studies, five [12,13,17,18,19] combined physical therapy with BoNT-A treatment, while the remaining two studies used BoNT-A treatment alone [16] or supplemented it with a custom neck orthosis [15]. In particular, studies by Oleszek [12] and Sinn [19] included home stretching in addition to physical therapy, while one study [13] included paracetamol administration and various adjuvant therapies in addition to physical therapy. Limpaphayom et al. [17] implemented a structured physical therapy protocol consisting of guided sessions at least twice weekly, daily home stretching three or more times, and adjunctive measures such as muscle strengthening, kinesiotaping, and use of a total orthotic torticollis collar when necessary.

### 2.7. Injection Accuracy

In terms of injection accuracy, palpation and patient positioning techniques were employed in six studies [12,13,14,16,17,18], while electrical stimulation-assisted needle guidance was utilized in two studies [15,19].

### 2.8. Quality of Included Studies

The Joanna Briggs Institute (JBI) quality appraisal of the included case series showed generally good methodological quality, with most studies scoring positively on six or seven out of eight checklist items (Table 3). Common strengths across studies included clear inclusion criteria, valid identification of the condition, and thorough reporting of demographics, clinical information, and outcomes. However, some limitations were noted, such as lack of consecutive participant inclusion and incomplete follow-up, particularly in the studies by Oleszek [12] and Sinn [19]. The study by Han [14] met nearly all criteria but lacked consecutive participant inclusion. The case report by Sytsma [15] had some nonapplicable items due to its nature as a single case. Overall, the evidence supports the safety and effectiveness of botulinum toxin type A in treating congenital muscular torticollis, though further randomized controlled trials are needed for definitive conclusions.

According to the JBI critical appraisal for cohort studies, one study [17] clearly reported the inclusion criteria and follow-up period, and demonstrated methodological validity through the exclusion of major confounding factors and ethical approval (Table 4). However, the absence of a control group and certain limitations in outcome measurement methods should be considered as potential sources of bias.

The risk of bias assessment using the Cochrane risk of bias (ROB) 2.0 tool for the two included studies by Su [16] and Liu [18] is summarized in Figure 2. Both studies showed low risk of bias in domains related to deviations from intended interventions and measurement of the outcomes, reflecting adequate intervention fidelity and reliable outcome assessment. However, some concerns were noted in the domains assessing the randomization process, missing outcome data, and selection of reported results. These concerns indicate potential limitations such as incomplete allocation concealment, attrition bias, and risk of selective reporting. Overall, the assessments suggest moderate methodological quality requiring cautious interpretation of the results.

## 3. Discussion

### 3.1. Main Findings

This scoping review identified six clinical studies investigating the effectiveness and safety of BoNT-A for pediatric torticollis, comprising four case reports and two randomized controlled trials published between 2005 and 2023. The aggregated results demonstrate consistent clinical improvement in neck ROM, reduction in head tilt, and overall functional outcomes following BoNT-A injections in pediatric patients for whom conventional physical therapy was either insufficient or ineffective. Notably, the majority of included studies reported favorable outcomes after one or two sessions of BoNT-A injection targeted to the SCM and adjacent muscles, with a substantial proportion of participants achieving restored cervical function.

Regarding safety, mild and transient adverse events—such as local bruising, soreness, edema, and short-lived muscle weakness—were documented in a minority of cases, with no reports of severe or persistent complications. While the limited number of studies analyzed presents an inherent limitation in the generalizability of these findings, the observed safety profile aligns well with previously reported outcomes in the literature for BoNT-A use in pediatric neuromuscular disorders. This consistency reinforces that BoNT-A can be considered a minimally invasive adjunct therapy with low procedural risk, as supported by scoping reviews and clinical studies in similar patient populations [19].

From an implication standpoint, the available evidence suggests that BoNT-A injections offer clinical utility for pediatric torticollis patients who do not respond adequately to standard physical therapy. Incorporating BoNT-A into treatment protocols could reduce the likelihood of chronic musculoskeletal sequelae, facilitate earlier achievement of normal cervical motion, and potentially decrease the need for more invasive surgical interventions [19]. Nevertheless, the data highlight the importance of individualized dosing, careful muscle targeting, and ongoing monitoring for adverse events, given the relative paucity of large-scale randomized trials in the pediatric population.

In this review, six of the eight included studies employed palpation and patient positioning techniques to enhance BoNT-A injection accuracy. However, recent evidence highlights the importance of adjunctive methods, such as ultrasound and electromyography (EMG) guidance, to ensure precise administration and optimal therapeutic effect of botulinum toxin. Shipkov et al. [20] systematically reviewed injection techniques for cervical dystonia and reported that ultrasound-guided injections consistently achieve higher accuracy compared to palpation-based methods, while EMG further refines targeting by detecting dystonic muscle activity, particularly when anatomical distinctions are unclear or muscle function is altered.

These findings are consistent with other clinical reports showing that ultrasound and EMG guidance improve the precision of BoNT-A delivery, thereby enhancing treatment efficacy and safety. Therefore, for optimal therapeutic outcomes, clinicians are strongly advised to use these adjunctive tools alongside traditional techniques. Advanced imaging and electrophysiological guidance not only ensure accurate injection but also allow for potential dose reduction, reducing the risk of immunogenicity and improving overall patient satisfaction. Future research should focus on developing standardized protocols combining ultrasound and EMG to establish best-practice guidelines for BoNT-A injections in cervical dystonia and other movement disorders.

### 3.2. Limitations of the Review

The present scoping review has several limitations that should be acknowledged. Primarily, the included studies comprise a small number of randomized controlled trials along with several case reports and case series, limiting the overall strength of evidence and increasing susceptibility to bias. The diversity in study designs and variability in botulinum toxin dosages and injection protocols, as well as inconsistencies in follow-up durations, contribute to significant heterogeneity that impedes meta-analytic synthesis. Additionally, the relatively short follow-up durations restrict the evaluation of long-term efficacy and safety of BoNT-A in pediatric torticollis. A critical inherent challenge is the difficulty of conducting large-scale RCTs in pediatric populations due to ethical considerations, rarity of severe cases, and logistical constraints [21].

Given these challenges, alternative research approaches should be considered, such as leveraging real-world clinical data from pediatric clinics. Collecting observational data and employing robust analytical methods to compare treated and untreated populations could provide valuable insights while circumventing the practical difficulties of RCTs in children. Such real-world evidence can complement findings from small-scale trials and guide clinical decision-making [22,23].

Despite these limitations, this review’s strengths include the focused synthesis of evidence specifically on pediatric patients, addressing a notable gap, where previous research largely centers on adults. The comprehensive search strategy, extending to the most recent publications, enhances the relevance and currency of the findings.

Moreover, consistent reports of clinical improvement alongside a favorable safety profile reinforce BoNT-A as a promising adjunctive therapy in refractory pediatric torticollis, although long-term prospective studies with more homogeneous data are needed to better characterize sustained efficacy and safety outcomes. This warrants further investigation through innovative study designs and data sources.

### 3.3. Implications for Future Studies

Future studies investigating BoNT-A for pediatric torticollis should advance beyond preliminary findings by adopting comprehensive and standardized research frameworks. No ongoing clinical trials investigating BoNT-A for pediatric torticollis are currently registered on ClinicalTrials.gov. Retrospective clinical studies have demonstrated promising safety and efficacy of BoNT-A in refractory pediatric cases; however, prospective studies with larger sample sizes and standardized protocols are needed to validate these findings and optimize treatment parameters. Although one study [19] examined a relatively large cohort from a single center (e.g., 134 children), multicenter efforts remain important to generalize the findings and account for potential variability across populations. Standardization of treatment protocols—including dosing regimens, injection sites, and follow-up periods—is critical to enable comparability across studies and to identify optimal therapeutic parameters [24]. Incorporating multimodal outcome measures that combine objective assessments (e.g., cervical range of motion, muscle ultrasound imaging) with patient-centered metrics (e.g., quality of life, caregiver burden) will deepen understanding of BoNT-A’s clinical impact [25].

Beyond conventional randomized controlled trials, alternative research frameworks such as pragmatic clinical trials and registry-based observational studies should be utilized to better reflect real-world treatment practices and to capture long-term safety outcomes. Such approaches may help to address the ethical and logistical challenges often encountered in pediatric trials, while still generating high-quality evidence through the application of advanced statistical methods, including propensity score analyses and machine learning techniques. Furthermore, investigating the integration of adjunctive interventions—such as physiotherapy or orthotic management—alongside BoNT-A may yield valuable insights into potential synergistic effects and contribute to optimizing therapeutic outcomes [17]. Ultimately, a multidisciplinary research agenda that combines clinical expertise with methodological rigor and technological innovation will be essential to establish evidence-based standards and advance the quality of care for children with torticollis [26]. The use of emerging technologies, particularly ultrasound-guided injection techniques, also warrants further study to improve procedural accuracy and minimize adverse events.

## 4. Conclusions

This systematic review comprehensively evaluated the clinical efficacy and safety of BoNT-A in infants and children with torticollis. Across the six included studies, BoNT-A treatment consistently demonstrated improvements in cervical range of motion, correction of head tilt, and functional recovery. Most reported adverse effects were mild and transient, such as localized bruising, swelling, or minor neck weakness, supporting the favorable safety profile of BoNT-A as an adjunct to conventional physical therapies in patients unresponsive to conservative management.

Nevertheless, there are notable gaps in knowledge regarding optimal dosing regimens, injection sites, standardized outcome measurements, and long-term effects due to variability in study methodologies and limited follow-up durations. Given the ethical and logistical challenges of conducting large randomized controlled trials in pediatric populations, future research should focus on multicenter collaborations, standardized clinical protocols, longer monitoring periods, and incorporation of patient-centered outcomes. Additionally, integrating real-world evidence and advanced imaging techniques may enhance understanding and optimize the use of BoNT-A in managing pediatric torticollis.

## 5. Materials and Methods

### 5.1. Study Design and Registration

The study selection process for this scoping review was conducted in accordance with the PRISMA Extension for Scoping Reviews (PRISMA-ScR) guidelines [27]. A complete PRISMA-ScR checklist is presented in Appendix A. The study protocol was preregistered with the Open Science Framework (OSF) (https://doi.org/10.17605/OSF.IO/AXWDP, accessed on 31 August 2025).

### 5.2. Data Sources and Searches

A literature search encompassing twelve electronic databases was undertaken: PubMed, EMBASE, Cochrane Central Register of Controlled Trials, CINAHL Plus, Korean databases (ScienceON, Korean traditional knowledge portal, Korea Citation Index, Research Information Sharing Service, OASIS, and Korean Medical database), and Chinese databases (China National Knowledge Infrastructure (CNKI) and Wanfang). Relevant papers published up until June 2025 were included in the review.

The search terms used were ((“Botulinum Toxins, Type A” OR “botulinum toxin type A” OR “Botulinum toxin A) AND (“pediatric torticollis” OR “congenital muscular torticollis” OR “Congenital torticollis”) AND (“clinical trial” OR “randomized controlled trial” OR “controlled clinical trial” OR “case report”)). The search terms for each database are included in Appendix A.

### 5.3. Study Selection

After the initial search, duplicate records were removed, and two independent reviewers (N.-Y.K and S.-H.S.) screened the titles and abstracts. Full texts of potentially eligible studies were then assessed to determine final inclusion. Eligible studies included clinical trials, controlled studies, or case reports that investigated the use of BoNT-A in children and adolescents under 20 years of age with congenital muscular torticollis. Non-clinical research, experimental animal studies, and review articles were excluded. Studies were required to report at least one quantitative outcome, such as muscle tone, ROM, or symptom improvement. Disagreements between reviewers were resolved through discussion and, when necessary, adjudication by a third reviewer (H.-K.S.) to minimize selection bias.

#### 5.3.1. Inclusion Criteria

The inclusion criteria were as follows: pediatric patients younger than 20 years who were clinically diagnosed with congenital muscular torticollis; patients who showed insufficient response to conservative management, including physical therapy; and studies that involved clinical trials, controlled studies, or case reports reporting at least one quantitative outcome measure related to symptom improvement, such as cervical range of motion or correction of head tilt. In addition, only participants who received BoNT-A treatment were included.

#### 5.3.2. Exclusion Criteria

The exclusion criteria were as follows: studies involving adult patients (≥20 years) or animal models; non-clinical articles such as reviews, editorials, or laboratory-only experimental studies; studies lacking quantitative clinical outcomes or sufficient data on treatment effects; and duplicate publications or studies that did not clearly specify the inclusion of the target patient population.

### 5.4. Data Extraction

Data extraction was independently conducted by two reviewers (N.-Y.K and S.-H.S.) using a pre-designed standardized form to ensure consistency and accuracy, following established systematic review methodologies. Extracted information included study characteristics (author, year, country), study design (RCT, case report, observational), participant demographics (sample size, age, sex), details of BoNT-A intervention (dose, volume, injection sites, number of treatment sessions, administration route, product used), co-interventions, outcome measures, and reported adverse events. Discrepancies in extracted data between reviewers were resolved through discussion and consensus or involvement of a third reviewer (H.-K.S.) when necessary to minimize bias and enhance data reliability. Efforts were made to retrieve missing or unclear information by contacting study authors if possible.

### 5.5. Assessment of the Risk of Bias in the Included Studies

To evaluate the methodological quality and risk of bias in the included studies, we employed the Cochrane ROB tool version 2.0 for RCTs [28]. Two independent reviewers assessed bias domains including randomization process, deviations from intended interventions, missing outcome data, measurement of the outcome, and selection of reported results. For case reports and case series, the JBI Critical Appraisal Checklist for Case Reports was utilized [29]. This checklist assesses domains such as clear description of patient demographics, clinical history, intervention details, outcomes, adverse events, and follow-up adequacy. Discrepancies were resolved through discussion and consensus with a third reviewer when necessary to ensure objectivity. The methodological quality of cohort studies included in this review was critically appraised using the JBI Critical Appraisal Checklist for Cohort Studies [29], which comprises 11 items assessing key domains such as selection bias, measurement of exposures and outcomes, confounding factors, follow-up completeness, and appropriateness of statistical analyses. Two independent reviewers performed the appraisal, and any disagreements were resolved by consensus. This rigorous evaluation ensured that the included cohort studies met methodological standards necessary to minimize bias and enhance the validity of findings.

### 5.6. Effectiveness and Safety Assessment

Effectiveness outcomes across the included studies were primarily evaluated using standardized clinical measures such as ROM, degree of head tilt, muscle thickness assessed by ultrasound, and patient- or caregiver-reported satisfaction scores.

Safety data were collected for adverse events occurring in the BVA group throughout the study period of the included clinical trials. Reported adverse events were categorized according to the Common Terminology Criteria for Adverse Events (CTCAE) [30]; (1) Mild (Grade 1): Asymptomatic or presenting with mild symptoms; only clinical or diagnostic observations are required; no intervention indicated. (2) Moderate (Grade 2): Minimal, local, or noninvasive intervention indicated; may limit age-appropriate instrumental activities of daily living. (3) Severe (Grade 3): Medically significant but not immediately life-threatening; hospitalization or prolongation of hospitalization may be required; disabling; limits self-care activities of daily living. (4) Life-threatening (Grade 4): Immediate risk of death; urgent intervention required. (5) Death (Grade 5): Adverse event results in death.

## Figures and Tables

**Figure 1 toxins-17-00543-f001:**
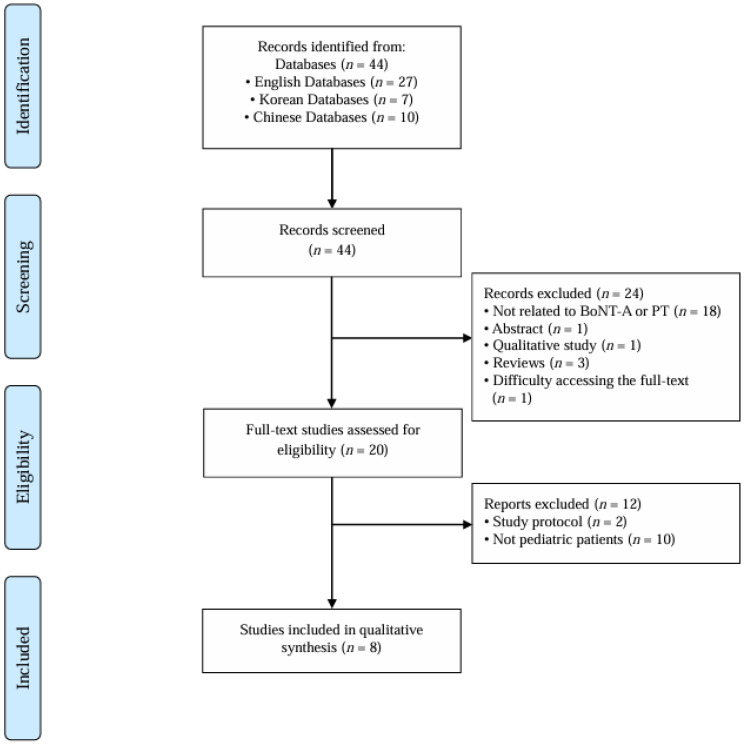
Overview of the study selection procedure according to the inclusion and exclusion criteria. BoNT-A: botulinum toxin type A; PT: pediatric torticollis.

**Figure 2 toxins-17-00543-f002:**
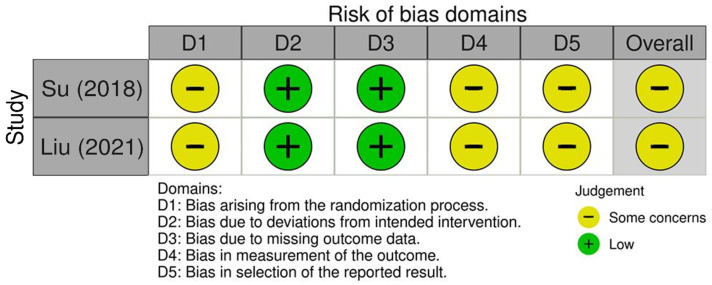
Risk of bias assessment for the included studies using the Cochrane risk of bias assessment 2.0 tool [16,18].

**Table 1 toxins-17-00543-t001:** Characteristics of included clinical studies.

	First Author (Year, Countries)	Study Design	Number of Patients (Age, M/F)	Botulinum Toxin (Dose, Volume, Sessions, Injection Points, Injection Route)	Outcome Measure	Main Result *	Adverse Events	Co-Intervention	Injection Accuracy
1	Oleszek (2005) [12] USA	Case report	*n* = 27 Age: 6~18 months (mean 10.1 moths) M/F: n.r.	1. Dose: 100 U/0.5 mL 2. Volume: based on age and muscle size 3. 1 session per 27 patients, and 2 sessions for 3 patients 4. SCM or upper trapezius muscle 5. IM	ROM (cervical rotation and active head tilt)	Improved (74%)	7% transient AEs (mild dysphasia of 12 mos, neck weakness of 10 mos)	Physical therapy and home program	Palpation and patient positioning
2	Joyce (2005) [13] New Zealand	Case report	*n* = 15 Age: 3–17 months (mean 7.6 months) M/F: 11/4	1. Dose: 25~50 IU (mean 33.3 IU) based on body weight 2. Volume: 1 mL 3. 1 session per 27 patients, and 2 sessions for 3 patients 4. SCM 5. IM	1. ROM (active and passive ranges of lateral rotation and side flexion of the neck) 2. Patient satisfaction	1. Improved 2. Highest in BoNT-A	Bruising of the neck (1), sore neck (1)	Physiotherapy * stretching exercise Anesthesia and paracetamol for discomfort (9 for active counter positioning, 5 for orthotic helmet, 1 for soft collar, 1 for craniosacral osteopathy)	Palpation and patient positioning
3	Han (2011) [14] Republic of Korea	Case report	*n* = 52 (E) *n* = 46 age: 13–90 days (mean 32.4 ± 17.1 days) M/F: 29/17 (C) *n* = 6 age: 16–90 days (mean 47.7 ± 33.3 days) M/F: 3/3	1. Dose: 2~4 U/kg (diluted to 2 cc with 0.9% saline) 2. Volume: 2 cc 3. 1 session (repeat if needed) 4. 1~2 points of SCM 5. IM	1. SCM thickness difference (US) 2. Duration and number of stretching sessions 3. Neck ROM	88.5% (*n* = 46) recovered with stretching alone in 1–6 weeks 11.5% (*n* = 6) required BoNT-A and/or surgery Larger SCM thickness difference correlated with longer therapy and need for additional interventions	Mild dysphagia (7), neck weakness (7)	All patients: manual stretching + therapeutic ultrasound, massage, home exercise Group 2: ±BoNT-A, ±surgery	Palpation and patient positioning
4	Sytsma (2016) [15] USA	Case report	*n* = 1 Age: 32 months M/F:1/0	1. Dose: 50 U BoNT-A in 0.5 mL of normal saline for 1st Tx, 100 units BoNT-A in 1 mL of normal saline for 2nd Tx 2. Volume: n.r. 3. 2 sessions 4. SCM, upper trapezius 5. IM	ROM (head tilt, shoulder elevation with head rotation)	Improved (only 10° intermittent head tilt and no tilt or shoulder elevation with leftward head rotation to 90°)	No AEs reported	Custom neck orthosis (8–10 h/day for 3 months)	Needle guidance using electrical stimulation
5	Su (2018) [16] China	RCT	*n* = 18 Age: 19~100 days (mean 52 days) M/F: 11/7	1. Dose: 19.30 ± 3.88 U (100 U dissolved in 2 mL normal saline) 2. 1 mL, 4 U/kg 3. 1 session (*n* = 26), 2 sessions (*n* = 4) after 6 months 4. 1 point of SCM in A group, 3 points of SCM in B group 5. IM	1. Head tilt and ROM 2. Thickness of the lower segment of SCM by ultrasound 3. Length of the SCM	1. Improved (*n* = 18) 2. No statistically significant difference between 2 groups after Tx 3. Significant difference between 2 groups 3 months after treatment (*p* < 0.05)	Slight weakness in sucking on the day of injection (*n* = 1), local subcutaneous bruising (*n* = 3)	None	Palpation and patient positioning
6	Limpaphayom (2019) [17] USA and Thailand	Cohort study	*n* = 39 Age: 6.5–27.6 months (mean 14 months) M/F: 20/19	1. Dose: 2–3 U/kg 2. Volume: 0.5 mL (100 U reconstituted in saline) 3. 1–4 (median 2; repeat every ~20–28 weeks if needed) 4. 2–3 points of SCM 5. IM	1. Head tilt 2. Neck rotation (ROM) 3. Caregiver satisfaction	1. Significant improvement: head tilt from 18.7° → 1.7° 2. ROM from 56° → 86° (*p* < 0.001) 3. No child required tendon lengthening surgery; high caregiver satisfaction (89%)	n.r.	Structured physical therapy protocol: guided PT ≥ 2 × /week, home stretching ≥ 3 × /day, strengthening, kinesiotaping, TOT collar if needed; continued post-BTX PT 6–8 wks	Palpation and patient positioning
7	Liu (2021) [18] China	RCT	*n* = 23 (E) age: 5~16 months (mean 7.6 ± 2.2 months) M/F: 8/3 (C) age: 6~14 months (mean 8.3 ± 3.6 months) M/F: 6/6	1. Dose: 100 U (dissolved in 1.0 mL of sterile 0.9% sodium chloride injection) 2. Volume: 1 mL (3 U/kg) 3. 1 session 4. 3 points of SCM 5. IM with Lidocaine ointment	1. Degree of head tilt 2. ROM (neck rotation)	1. Significant improvements in head tilt (*p* < 0.05) 2. Significant improvements in neck rotation (*p* < 0.05)	Subcutaneous bruising (*n* = 1) or subcutaneous edema (*n* = 1) in 18.18% (2/11) in intervention; in control group, subcutaneous edema (*n* = 1) in 8.33% (1/12) (*p* > 0.05)	Physical therapy (10 sessions for 8~10 weeks)	Palpation and patient positioning
8	Sinn (2023) [19] USA	Case report	*n* = 134 Age: 6–24 months (mean 11.9 ± 3.9 months) M/F: 82/52	1. Dose: 15~30 U per site based on clinical judgment 2. Volume: n.r. 3. 1 session (*n* = 122), 2 sessions (*n* = 12) 4. SCM, upper trapezius, scalenes 5. IM	ROM (active lateral flexion and active cervical rotation)	Improved (61%, *n* = 82)	No AEs reported	Physical therapy and home stretching	Needle guidance using electrical stimulation

* For case reports or single-arm studies (*n* = 1), the term “improved” was used when authors reported post-treatment improvements in clinical outcomes, although no statistical analysis was conducted. For RCTs and CCTs, statistical significance (e.g., *p*-values) was reported when available. AEs: adverse events; BoNT-A: botulinum toxin type A; F: Female; IM: intramuscular (injection); IU: International Unit; M: Male; n.r.: not reported; RCT: randomized controlled trial; ROM: range of motion; SCM: sternocleidomastoid muscle.

**Table 2 toxins-17-00543-t002:** Detailed utilization of botulinum toxin type A.

First Author, Year	Dose	Volume		Sessions	Injection Points	Route of Administration	Products
		Volume per Session (mL)	Volume for Entire Treatment (mL)				
Oleszek (2005) [12]	100 U/0.5 mL	based on age and muscle size		1 session per 27 patients, and 2 sessions for 3 patients	SCM or upper trapezius muscle	IM	BoNT-A (Allergan Inc., Irvine, CA, USA)
Joyce (2005) [13]	25~50 IU (mean 33.3 IU)	1 mL		1 session with general anesthesia	3~4 points of SCM	IM	BoNT-A (Allergan Inc., Irvine, CA, USA)
Han (2011) [14]	2~4 U/kg	2 cc		1 session (repeat if needed)	1~2 points of SCM	IM	BoNT-A (Allergan Inc., Irvine, CA, USA)
Sytsma (2016) [15]	50 U	0.5 mL	1 mL	2 sessions	SCM, upper trapezius	IM	BoNT-A (Allergan Inc., Irvine, CA, USA)
Su (2018) [16]	19.30 ± 3.88 U	1 mL, 4 U/kg		1 session (*n* = 26), 2 sessions (*n* = 4) after 6 months	SCM	IM	BoNT-A (Hengli^®^, Lanzhou Institute of Biological Products Co., Ltd., Lanzhou, China)
Limpaphayom (2019) [17]	2–3 U/kg	0.5 mL (3 U/kg)		1–4 sessions	2–3 points of SCM	IM	BT-A (Allergan Inc., Irvine, CA, USA)
Liu (2021) [18]	100 U	1 mL		1 session with Lidocaine ointment	3 points of SCM	IM	BoNT-A (Hengli^®^, Lanzhou Institute of Biological Products Co., Ltd., Lanzhou, China)
Sinn (2023) [19]	15–30 U	n.r.		2 sessions	Ipsilateral sternocleidomastoid, upper trapezius, and scalene muscles	IM	n.r.

BoNT-A: botulinum toxin type A, IM: intramuscular (injection), IU: International Unit, n.r.: not reported, SCM: sternocleidomastoid muscle, U: Unit.

**Table 3 toxins-17-00543-t003:** Risk of bias assessment for the included studies based on the Joanna Briggs Institute Critical Appraisal Checklist for Case Reports.

First Author, Year	Q1	Q2	Q3	Q4	Q5	Q6	Q7	Q8
Oleszek (2005) [12]	Yes	No	Yes	No	No	Yes	Yes	Yes
Joyce (2005) [13]	Yes	Yes	Yes	No	Yes	Yes	Yes	Yes
Sytsma (2016) [15]	Yes	Yes	Yes	NA	NA	Yes	Yes	Yes
Sinn (2023) [19]	Yes	No	Yes	No	No	Yes	Yes	Yes
Han (2011) [14]	Yes	Yes	Yes	Yes	Yes	Yes	No	Yes

**Table 4 toxins-17-00543-t004:** Risk of bias assessment for the included studies based on the JBI Critical Appraisal Checklist for Cohort Studies.

First Author, Year	Q1	Q2	Q3	Q4	Q5	Q6	Q7	Q8	Q9	Q10	Q11
Limpaphayom (2019) [17]	Yes	Yes	Yes	No	No	Yes	Yes	Yes	No	No	Yes

## Data Availability

No new data were created or analyzed in this study. Data sharing is not applicable to this article.

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
