# Peer review of "Botulinum Toxin Type A for Pediatric Torticollis: A Review of Clinical Research"

_toxins, 2025, doi:10.3390/toxins17110543_

Round 1
Reviewer 1 Report
Comments and Suggestions for Authors
This is a necessary study since the related series are scares.
I would have the following recommendations for corrections:
- The authors talk about inclusion and exclusion criteria - please, list both of them explicitly.
- The authors should add some more discussion and details about the technique of injection. The technique should be clearly described. Do the authors use any additional methods when injectinh - for example US, EMG or other? In this sense a recent review could be used and sited: Comparing Injection Methods of Botulinum Toxin A for Cervical Dystonia: A Systematic Review. Shipkov H, Uchikov P, Imran A, Ul Hassan Z, Grozdev I, Kraev K, Kraeva M, Koleva N, Bozhkova M, Karamitev S.Life (Basel). 2025 Jun 6;15(6):920. doi: 10.3390/life15060920.PMID: 40566572 Free PMC article. Review. The overall impression is of an interesting manuscript with definite clinical significance.
Author Response
Thank you to the reviewers for their careful evaluation of our manuscript. Please refer to the attached file for our responses to each comment.

Reviewer 2 Report
Comments and Suggestions for Authors
The work is original, references are appropriate. BT A could be a valuable therapeutic option for pediatric torticollis but it is a scoping review with only six heterogeneous cases withoutstandardization: variability in dosing, injection protocol, follow up durations. Long-term prospective studies are needed and more homogeneous data. Could be useful to search clinical trials ongoing on clinical trial.gov
Author Response

(The authors gave the same response as above.)

Reviewer 3 Report
Comments and Suggestions for Authors
toxins-3877995
This manuscript deals with an important subject, namely the treatment of children, with specific muscular conditions, with BoNT-A.
However, review of the manuscript took considerable time as many issues were identified in the content. Significant work is required to improve the content, in particular in relation to missing studies.
My comments are as follow:
I recommend that “Scoping” be removed from the title and content – this has no relevant meaning here
The correct abbreviation should be used BoNT-A
Lines 10, 43 BoNT-A is a blocking agent, not an inhibitory agent
Introduction
For citation [1], why have the authors used the 2018 Clinical practice Guideline instead of the 2024 version?
Sargent, B., Coulter, C., Cannoy, J., & Kaplan, S. L. (2024). Physical Therapy Management of Congenital Muscular Torticollis: A 2024 Evidence-Based Clinical Practice Guideline From the American Physical Therapy Association Academy of Pediatric Physical Therapy. Pediatr Phys Ther, 36(4), 370–421. https://doi.org/10.1097/PEP.0000000000001114
Use of the 2024 guideline may require changes to the manuscript content – to be determined by the authors.
Citations [7] and [8] are too old. More recent citations should be used.
I could not locate citation [8] by title, DOI or directly in the journal
Citation [9] is related to patients overall and not to a pediatric population. Because children’s neuromuscular systems are under development, the side effect profile for BoNT-A in that population may be different. The authors should add a commentary to that effect.
Line 54 is in error
Methods
Lines 261-264 The searches did not include the term “congenital muscular torticollis”. The authors should comment.
Results
Citations 11-16 should be in date order
The title and DOI for citation [11] are incorrect
The page details for citation [12] are incorrect
I could not locate citation [16] by title, DOI or directly in the journal
It is not possible to check the details for citations [15] and [16] as they are in Chinese publications.
The following publications should have been included in the evaluation or their omissions should be justified by the authors:
Jo, H. S., Kang, Y. K., Paik, K. W., Kim, D. H., Hwang, M., R., & Kim, K. H. (2002). Local Botulinum Toxin Type A Injection for the Management of Congenital Muscular Torticollis. Ann Rehabil Med, 26(6), 699–703.
Do, T. T. (2006). Congenital muscular torticollis: current concepts and review of treatment. Curr Opin Pediatr, 18(1), 26–29. https://doi.org/10.1097/01.mop.0000192520.48411.fa
Han, J. D., Kim, S. H., Lee, S. J., Park, M. C., & Yim, S. Y. (2011). The thickness of the sternocleidomastoid muscle as a prognostic factor for congenital muscular torticollis. Ann Rehabil Med, 35(3), 361–368. https://doi.org/10.5535/arm.2011.35.3.361
Gunduz, A., Korkmaz, B., & Kiziltan, M. E. (2014). Effective treatment of congenital muscular torticollis using botulinum toxin. J Craniofac Surg, 25(5), 1935. https://doi.org/10.1097/SCS.0000000000001123
Sinn, C. N., & Rinaldi, R. J. (2016). Treatment with Botulinum Toxin Type A in Infants with Refractory Congenital Muscular Torticollis: A 10-Year Retrospective Study. Pm r, 8(9S), S152–S153. https://doi.org/10.1016/j.pmrj.2016.07.023
Jiang, B., Zu, W., Xu, J., Xiong, Z., Zhang, Y., Gao, S., Ge, S., & Zhang, L. (2018). Botulinum toxin type A relieves sternocleidomastoid muscle fibrosis in congenital muscular torticollis. Int J Biol Macromol, 112, 1014–1020. https://doi.org/10.1016/j.ijbiomac.2018.02.077
Limpaphayom, N., Kohan, E., Huser, A., Michalska-Flynn, M., Stewart, S., & Dobbs, M. B. (2018). Use of Combined Botulinum Toxin and Physical Therapy for Treatment Resistant Congenital Muscular Torticollis. J Pediatr Orthop, 39(5), e343–e348. https://doi.org/10.1097/bpo.0000000000001302
Qiu, X., Cui, Z., Tang, G., Deng, H., Xiong, Z., Han, S., & Tang, S. (2020). The Effectiveness and Safety of Botulinum Toxin Injections for the Treatment of Congenital Muscular Torticollis. J Craniofac Surg, 31(8), 2160–2166. https://doi.org/10.1097/SCS.0000000000006652
Martinez, B. M., Stella, J., Fortoul, M. C., Oualid, C., Kim, J., & Kamel, G. (2025). A Retrospective Review: Treatment of Congenital Muscular Torticollis. Ann Plast Surg. https://doi.org/10.1097/SAP.0000000000004436
These were all identified by searching for “congenital muscular torticollis in the Title.
I have not checked the details included in Table 1 against each publication cited. However, this table may change depending on the authors responses to my point about missing study citations above. Given issues with some of the citation details, the authors should be encouraged to check the details in Table 1 fully and report those checks to the Editors.
Line 87, Table 1 and Table 2 There is no such thing as an International Unit (IU) of botulinum toxin. Each product has units specific to that product alone. Therefore when units are quoted, the product must also be stated.
In Table 2, the product used in the Sytsma citation [13] is in the title of the publication – ONA from Allergan
Line 106 What is “The evaluation index for Botox treatment….” ? A citation should be provided.
Sections 2.4 and 2.6 Here, the authors have started to use the term “Botox”, but this was not the only product used in the cited studies [11-16]
Line 120 Again, the term “Botox” has been used for the Chinese BoNT-A product, which is incorrect
Line 136 “Table X” should be Table 3. Also, the first part of the table shows data for more than citations [15] and [16].
A citation in the legend for Table 3 should be included
Why were only citations [15] and [16] included in the Risk of Bias Domains analysis?
Discussion
Lines 167 & 173 Citation [17] relates to BoNT-A use in Mohs surgery cases. These are completely unrelated to the pediatric population discussed in the manuscript. This citation must be changed.
Line 175 Again, citation [18] delas with treatment of drooling, which is a condition unrelated to the torticollis discussed in the manuscript. This should also be changed.
Line 176 et seq I disagree with this statement. There are over 1000 publications dealing with children and the use of BoNT-A for their therapy, most notably in the treatment of cerebral palsy.
Lines 209-210 “…rarity and heterogeneity of this pediatric condition….” To argue against this statement, citation [14] examined 134 children from one center alone.
Line 225 Citation [24] is unrelated to the information discussed in this sentence and should be changed.
Line 230 Citation [25] should be moved to the previous sentence.
Author Response

(The authors gave the same response as above.)

Round 2
Reviewer 3 Report
Comments and Suggestions for Authors
All my review comments have been addressed in the revised version